# Potential Mechanisms for How Long-Term Physical Activity May Reduce Insulin Resistance

**DOI:** 10.3390/metabo12030208

**Published:** 2022-02-25

**Authors:** Sindre Lee-Ødegård, Thomas Olsen, Frode Norheim, Christian Andre Drevon, Kåre Inge Birkeland

**Affiliations:** 1Department of Clinical Medicine, Faculty of Medicine, University of Oslo, 0372 Oslo, Norway; sindre.lee@medisin.uio.no; 2Department of Nutrition, Institute of Basic Medical Sciences, Faculty of Medicine, University of Oslo, 0372 Oslo, Norway; thomas.olsen@medisin.uio.no (T.O.); frode.norheim@medisin.uio.no (F.N.); c.a.drevon@medisin.uio.no (C.A.D.); 3Vitas Ltd. Analytical Services, Oslo Science Park, 0349 Oslo, Norway

**Keywords:** insulin resistance, exercise, long-term, adipose tissue, muscle, liver

## Abstract

Insulin became available for the treatment of patients with diabetes 100 years ago, and soon thereafter it became evident that the biological response to its actions differed markedly between individuals. This prompted extensive research into insulin action and resistance (IR), resulting in the universally agreed fact that IR is a core finding in patients with type 2 diabetes mellitus (T2DM). T2DM is the most prevalent form of diabetes, reaching epidemic proportions worldwide. Physical activity (PA) has the potential of improving IR and is, therefore, a cornerstone in the prevention and treatment of T2DM. Whereas most research has focused on the acute effects of PA, less is known about the effects of long-term PA on IR. Here, we describe a model of potential mechanisms behind reduced IR after long-term PA to guide further mechanistic investigations and to tailor PA interventions in the therapy of T2DM. The development of such interventions requires knowledge of normal glucose metabolism, and we briefly summarize an integrated physiological perspective on IR. We then describe the effects of long-term PA on signaling molecules involved in cellular responses to insulin, tissue-specific functions, and whole-body IR.

## 1. Background

Diabetes is one of the major medical challenges of modern times, estimated to affect 450 million people in 2017, and increasing to almost 700 million by 2045 [1]. Diabetes is defined as chronic hyperglycemia and is diagnosed by measuring a high glucose concentration in blood. There are different types of diabetes with different etiologies, the two major ones being type 1, caused by insulin deficiency due to autoimmune β-cell failure, and type 2 diabetes mellitus (T2DM) characterized by insulin resistance (IR) and relative insulin deficiency [2]. The term “diabetes (Greek for passing through) Mellitus (Latin for sweet)” comes from the fact that blood glucose levels exceed the kidneys’ reabsorption capacity, resulting in glucosuria. The high concentration of glucose in the pre-urine pulls along large amounts of water by osmosis and promotes significant diuresis. Hyperglycemia is the main cause of the microvascular complications of diabetes (retinopathy, nephropathy, and neuropathy) and can also play a major role in the increased propensity to atherosclerotic vascular diseases [3]. These complications are associated with reduced quality of life and premature mortality and are extremely expensive for society [3].

T2DM is a heterogeneous metabolic disorder for which several novel sub-classifications have been proposed [2,4,5]. Genetic and environmental factors interact to predispose for the IR and β-cell failure that leads to manifest T2DM and some prospective human studies indicate that IR is the best predictor of future T2DM diagnosis [4,5]. In most subjects who develop T2DM, excess body fat and too little PA increase IR to a degree where the insulin-secreting capacity of the pancreatic β-cells are not capable of keeping up, glucose levels increase and T2DM is diagnosed [6,7,8].

### 1.1. IR

IR refers to a state in which insulin has reduced biological effects, primarily on the glucose transport into muscle and fat cells and on the suppression of endogenous glucose production, mainly from the liver [9]. In healthy persons, ingesting a carbohydrate-rich meal will lead to a subtle rise in blood glucose levels and incretin secretion from the gut that stimulates insulin secretion from pancreatic β-cells [9]. Blood glucose levels will then quickly restore normo-glycaemia due to insulin-stimulated glucose uptake in skeletal muscle (SkM), but also because of reduced glucose production in the liver [9]. With increasing SkM IR [10,11], eating a carbohydrate-rich meal will lead to an increased insulin secretory response, but blood glucose levels will still take longer to normalize or may remain high [9]. The molecular mechanisms behind IR involve a complex network of inter-tissue cross-talk and several indirect effects leading to impaired insulin signaling, as reviewed by Roden & Shulman, 2019 [9], but also summarized in the following sections.

T2DM is strongly associated with obesity [1]. However, only a subset of obese people develops T2DM, indicating the need also to examine other factors, such as a genetic predisposition towards T2DM [12]. Furthermore, obesity is largely a consequence of overconsumption of caloric dense food in combination with reduced physical activity (PA) [13,14]. The combination of genetic disposition, energy surplus, and reduced energy expenditure due to a sedentary lifestyle, is probably the main reason for developing obesity-related IR and β-cell failure [1].

### 1.2. Insulin Signalling and Contraction-Induced Glucose Uptake

When bound to insulin, the insulin receptor auto-phosphorylates and recruits substrates such as the insulin receptor substrate (IRS) and Src Homology 2 (SH2) domain-containing signaling molecule [14]. Phosphorylated IRS protein then binds to the P85 subunit of phosphatidylinositol-3-kinase (PI3K) and recruits the catalytic subunit P110 to activate PI3K. PI3K then activates phosphorylation of phosphatidylinositol to produce 3,4,5-phosphatidylinositol-triphosphate (PIP3) [14]. PIP3 may then activate the serine/threonine protein kinase Akt (protein kinase B) which subsequently activates a variety of substrates and mediates multiple insulin effects [14]. One important effect is monophosphorylating Akt substrate of 160 kDa (AS160) and the translocation of glucose transporter 4 (GLUT4) to the cell membrane to increase glucose uptake [14]. In SkM, muscle contraction may lead to increased glucose uptake independently of the insulin receptor [15]. Upstream signaling pathways leading to acute contraction-induced GLUT4 translocation may include AMP-activated protein kinase (AMPK), Ca2+/calmodulin-dependent protein kinase II (CaMKII), nitric oxide (NOS), and reactive oxygen species (ROS) [15]. AMPK and CaMKII seems to be key signaling kinases that increase GLUT4 expression after long-term PA, potentially via nuclear export of histone deacetylase (HDAC) 4/5, which results in histone hyperacetylation on the GLUT4 promoter and enhanced transcriptional activity [15].

### 1.3. PA vs. Diet

Both increased PA and dietary energy restriction may prevent or even reverse T2DM, reducing morbidity and mortality [16,17,18,19]. Our research group has been particularly interested in the effects of increased PA, without alteration of the diet.

A distinction must be made between the effects of acute exercise (one session) versus long-term exercise (increased PA for several weeks or months). One single bout of PA can normalize impaired insulin-stimulated glycogen synthesis in IR offspring of parents with T2DM [20], reverse the abnormal pattern of carbohydrate storage in IR individuals by improved SkM glucose uptake and glycogen synthesis, and reduced hepatic lipogenesis [21]. These effects are largely due to insulin-independent contraction-induced glucose uptake in SkM [20,22]. SkM contraction may increase glucose uptake 50-fold through distal insulin signaling effectors different from those activated by insulin [15,23]. This signaling pathway is preserved in IR SkM [15,23]. However, these effects are seen for only about 2–72 h after acute PA. On the other hand, long-term PA may substantially reduce basal (resting and fasting) IR, fat oxidation, and glucose tolerance [24,25,26,27,28]. Long-term PA may also improve body composition and promote many effects such as enhanced SkM mitochondria number and oxidative capacity, improve vascularization and cardiac output, reduce blood concentration of HbA1c and triglycerides, lower blood pressure, and reduce total and cardiovascular mortality [29,30,31,32,33,34,35,36]. Most of the available data do not allow a clear distinction between the effects of long-term PA per se, and the effects of weight loss on IR [37,38]. Although weight loss due to either dietary energy restriction or long-term PA may both reduce IR, greater effects are seen after PA-induced weight loss [37,38]. The greatest effect of long-term PA on IR is seen after interventions with combined endurance and strength exercise [26,27,28,38,39]. Endurance exercise means repeated and continuous movement of large SkM groups (e.g., by cycling, walking, running, swimming, skiing, etc.) and relies on anaerobic (high-intensity interval training) and aerobic (low-intensity continuous training) energy/ATP production [40]. Strength/resistance-focused PA includes exercise at high loads and/or with high metabolic stress, and relies on phosphocreatine and lactic anaerobic energy/ATP production [40].

### 1.4. Estimation of IR

The gold standard method for measuring IR is the hyper-insulinemic-euglycemic clamp [41]. The clamp is performed by infusing insulin to suppress endogenous glucose production and increase peripheral glucose disposal, and simultaneous infusion of glucose to maintain euglycemia (plasma glucose equal to 5.0 mmol/L). In the steady-state, glucose infusion equals glucose uptake, provided that endogenous glucose production is suppressed. The lower the need for glucose to maintain euglycemia, the higher the IR. Tracer-labeled glucose is used if the infused insulin is not sufficient to suppress endogenous glucose production.

### 1.5. Designing a PA Intervention for IR

In collaboration with the Norwegian School of Sport Sciences, our research group performed the hyper-insulinemic-euglycemic clamp before and after a 12-week PA intervention in the MyoGlu study. Three days passed from the last acute bout of PA to the clamp. MyoGlu is a controlled clinical PA trial designed to characterize in-depth the effects of intensive PA intervention on IR in men [26]. The MyoGlu design included an acute bicycle test performed before and after the 12-week PA intervention. The PA included endurance training by two-cycle sessions (60 min each) and two whole-body resistance-training sessions (60 min each) each week. The resistance training focused on large muscle groups using exercises such as leg press, leg curl, chest press, cable pull-down, shoulder press, seated rowing, abdominal crunches, and back extension. PA followed a linear progression in volume through the 12 weeks for both strength and endurance exercises. For resistance exercises, the first 3 weeks included weights that could be lifted a maximum of 12 times, which progressed to heavier weights that could only be lifted 10 times in weeks 4 to 8, and then further to 8 repetitions in weeks 9 to 12. Abdominal crunches and back extensions were completed using weights lifted 12–20 times for the whole intervention, as recommended for core exercises. For endurance training, in weeks 1 to 3, cycle intervals at >90% of maximum heart rate were performed during session 1, and 6 intervals at 85% of maximum heart rate were completed in session 2. During weeks 2 to 5, the load was increased to 4 intervals during session 1 and 7 intervals during session 2. During weeks 6 to12, 5 intervals were completed during session 1 and 10 during session 2. All PA sessions were supervised by trained personnel.

## 2. Effects of Long-Term PA on IR

PA is a cornerstone in the prevention and treatment of T2DM. However, the mechanisms explaining the positive effects of PA on IR are incompletely understood. Some early reports demonstrated that only one exercise session may normalize glycogen synthesis and IR in T2DM patients and reduce liver fat content and de novo lipogenesis [21]. Contraction of SkM cells may also promote GLUT4-translocation and glucose uptake independent of insulin, both in insulin-sensitive persons and in patients with T2DM [42]. Hence, acute PA may bypass IR to increase SkM glucose uptake, which of course is of high clinical relevance. Notably, not all forms of PA may increase glucose uptake, such as eccentric exercise, where the muscle is stretched while stimulated to contract [43]. Furthermore, other effects of insulin are also enhanced by acute PA such as the stimulation of amino acid uptake [44] and protein synthesis [45]. However, acute effects of PA are outside the scope of this article and we refer to other recent reviews on this topic [46].

Although extensive research has been performed on acute PA effects on glucose uptake, less attention has been paid to the long-term effects of PA on IR. Perhaps this is because performing a well-controlled long-term PA intervention is both expensive and technically challenging, e.g., designing the study requires an depth understanding of exercise physiology, trained personnel are needed to supervise exercise sessions, and one has to distinguish the effects of long-term PA from the acute effects of the last bout of exercise in the intervention [26]. Emerging evidence points to the combined effect of both endurance and strength exercise as most efficient in reducing IR in a long-term PA intervention [26,27,28,38,39]. The following sections focus on the effects of long-term PA on different factors influencing IR.

### 2.1. SkM

#### 2.1.1. Ectopic Lipids in SkM

Early human research on IR focused on the SkM because most of the glucose in the non-fasting individual is utilized by SkM. Glucose has three potential fates in SkM. It can be incorporated into glycogen, or metabolized by oxidative or non-oxidative glycolysis. Measurements of intramuscular glycogen content have revealed that patients with T2DM may have only 50% increased SkM glycogen content during infusion of glucose, as compared to healthy subjects [47]. However, impaired SkM glycogen synthesis is not the primary defect in T2DM because intramuscular glucose, glucose-6-phosphate, and UDP-glucose are all low in T2DM, indicating that glucose transport must be the rate-limiting step in SkM IR [48]. The search for the cause of reduced SkM glucose uptake led to the “lipo-centric” view of IR because of strong correlations observed between SkM fat content (but not body mass index) and IR [49,50,51]. When lipids are infused together with insulin, SkM PI3k activity does not respond to insulin [52], believed to be caused by an accumulation of *sn*1,2-DAG activating protein kinase C (PKC) theta, which phosphorylates the insulin receptor and blocks recruiting insulin receptor substrate 1, thereby stopping GLUT4 translocation [53,54,55].

Paradoxically, endurance-trained athletes also often display high lipid content in SkM, despite being very insulin sensitive [56]. This observation has been referred to as “the athlete’s paradox”. However, new tissue staining methods and electron microscopy studies have shown different subcellular locations of lipid droplets (e.g., subsarcolemmal vs. intramyofibrillar) [57], and differences between type 1 and type 2 muscle fibers may explain this paradox [58]. Thus, the amount of SkM lipids may be similar among patients with T2DM and athletes, but differences exist in the location and size of lipid droplets [57,58], as well as amounts and types of lipid droplet-bound perilipins [59]. Furthermore, after long-term PA a more “athlete-like” pattern of lipid droplets and perilipins are observed, even in subjects with IR [57,58]. Furthermore, athletes display great energy turnover and lipid oxidation capacity [60]. Thus, it may also be important to which degree SkM lipids are used, as opposed to being stored, thereby potentially inducing IR. From an evolutionary perspective, when humans were hunters/gatherers, periods of food abundance were followed by periods of food scarcity [61,62]. When food is scarce, higher SkM lipid content and IR represent conserved evolutionary mechanisms to conserve glucose (e. g. the brain, or the fetus in pregnant women) and for providing an easily accessible energy source [61,62]. However, modern people are often subjected to a surplus of foods. For the sedentary modern man, SkM lipids are thus not adequately used, glucose is preserved in abundance and the response to energetic surplus becomes pathological. This consequence is avoided by increasing PA. Recent data suggest that not only total SkM fat content but also lipid composition [63] is important for IR. A recent study showed that oleate may activate the SkM sn1,2-DAG–PKC theta pathway to induce IR and that palmitate favors ceramide accumulation and PKC zeta–PP2A activation, which may aggravate SkM IR [64]. The effects of PA on these conditions remain unknown.

#### 2.1.2. SkM Mitochondria

SkM ectopic lipid accumulation may promote a surplus of fatty acids and uptake by SkM and/or in combination with reduced fatty acids oxidation in SkM [9]. Fat oxidation occurs mainly in mitochondria [9]. Hence, much focus has been attributed to SkM mitochondrial biogenesis and mitochondrial content and function in the light of ectopic lipid accumulation and IR [9]. Subjects with T2DM may exhibit lower mitochondrial content and function (less protein/enzyme activity per mitochondrion) as compared to healthy controls [65], Small intermyofibrillar mitochondria (mitochondria that provide ATP for SkM contraction) with low respiratory-chain activity are also observed in patients with T2DM [66]. Furthermore, low respiratory-chain activity may be especially pronounced in subsarcolemmal mitochondria (mitochondria that provide ATP for insulin-signaling, etc.) [67]. Mitochondrial dysfunction may occur due to hyperglycemia- and hyperlipidemia-induced production of reactive oxygen species (ROS) seen in T2DM [68]. Lean and young offspring with severe IR of parents with T2DM may demonstrate greater SkM ectopic lipid content and lower mitochondrial ATP synthesis despite a comparable rate of lipolysis as compared to controls [54,69]. These subjects were also unable to increase ATP synthesis during a hyperinsulinemic-euglycemic clamp when compared to controls [70]. However, it is still unclear if SkM ectopic lipid accumulation is a consequence of mitochondrial dysfunction, or if already existing IR is the cause of mitochondrial dysfunction leading to lipid accumulation [54,69].

Long-term PA may increase SkM mitochondrial content substantially in parallel with reduced IR [71,72,73]. SkM mitochondria may increase in numbers, length and cross-sectional area after long-term PA [74]. Mitochondria in SkM are found mainly between myofibrils or below the sarcolemma [57,74], where the absolute volume of the intra-myofibrillar subpopulation is substantially larger than that of subsarcolemmal mitochondria [57,74]. The intermyofibrillar mitochondria contribute the most to increased SkM total mitochondrial content after long-term PA, although the relative volume increase is largest for subsarcolemmal mitochondria [74]. The first mitochondrial adaptation after long-term PA seems to be hypertrophy, followed by elongation [74]. Long-term PA also stimulates an increase in mitochondrial protein content, such as enzymes involved in β-oxidation, the citric acid cycle, and the electron transport chain [75]. The mitochondrial adaptations to exercise are mediated mainly through activation of AMP-activated kinase, calcium–calmodulin-activated kinases, p38 mitogen-activated protein kinases, and sirtuin [74]. These enzymes stimulate the master regulator peroxisome proliferator-activated receptor γ coactivator 1α [74]. The result is increased nuclear transcriptional activity of, e.g., estrogen-related receptors, myocyte enhancer factor-2, nuclear respiratory factors 1 and 2, and peroxisome proliferator-activated receptors [74]. In addition, nuclear-encoded mitochondrial transcription factor A may translocate into mitochondria [74]. The joint effect of these mediators probably explains most of the exercise-induced improvement in mitochondrial function [74]. Furthermore, Zhou et al. systematically evaluated the effect of long-term PA on the mitochondrial lifecycle [76], and observed that Drp1, a regulator of mitochondrial fission, is critical for regulating mitochondrial adaptations, reducing lipid accumulation in SkM, and reducing IR after long-term PA [76]. We also showed that a change in SkM composition of phosphatidylethanolamine and phosphatidylcholine after long-term PA may be related to both increased mitochondrial density and reduced IR [63]. Taken together, it may seem that the effect of long-term PA may counteract impaired mitochondrial function seen in IR and T2DM.

#### 2.1.3. SkM Inflammation

Much focus has been attributed to the adipose tissue in terms of inflammation-induced IR [71]. It has also been suggested that SkM inflammation may play a role in IR [9]. However, we performed a large-scale electron microscopy study, including thousands of SkM microsections, but did not observe any macrophages, lymphocytes, or granulocytes in or close to myocytes [57]. It may seem that inflammation in SkM is not related to the SkM itself but rather to immune cell infiltration in perimuscular adipose tissue [9].

The most responsive genes to long-term PA in SkM are related to the immune system and inflammatory pathways [77]. However, it should be noted that gene names are given largely for historical reasons, and many of these genes were first discovered in studies of the immune system [77]. The roles of these genes are certainly different in different cells and tissues [77]., e.g., we have shown that an increase in “immune genes” after long-term PA may be related to the SkM remodeling of the extracellular matrix [72,73,77].

#### 2.1.4. Myokines

SkM is recognized as an endocrine organ [78]. Proteins that are expressed and released from SkM are known as myokines [79]. Several hundred myokines have been identified [80,81,82], and myokines might promote many of the health effects of exercise through crosstalk between SkM and other organs, such as adipose tissue, liver, and pancreas [78]. Most studies on myokines have been focused on acute exercise [78] and have reported enhanced plasma concentrations of proteins such as angiopoietin-like 4 [83], IL-6 [84], and myostatin [85] immediately after exercise. The effects of myokines on glucose metabolism are generally lacking, but some data are available for IL-6 [78]. Upon muscle contraction, IL-6 is released into the circulation [86]. The amount of release of IL-6 depends on factors such as exercise intensity and pre-exercise glycogen content [87]. During and just after exercise, the main source of plasma Il-6 is SkM [88], which is related to the activation of c-Jun N-terminal kinase (JNK) and activator protein 1 (AP1) in myofibers [89]. Acute exercise-derived IL-6 may enhance glucose uptake, GLUT4 translocation, and glycogen synthesis in SkM via gp130Rb/IL-6 receptor agonist (IL-6Ra), which activates AMP-activated protein kinase (AMPK) and phosphoinositide 3-kinase [90,91,92]. Acute exercise-derived IL-6 may also affect insulin secretion by increasing glucagon-like peptide-1 secretion from both the intestine and pancreas [93]. However, long-term PA does not seem to influence IR via IL-6 [26], probably because exercise-induced adaptation in SkM includes enhanced pre-exercise glycogen content [93,94,95]. Very few studies have assessed the long-term effects of PA on myokines and IR. However, we have identified 289 potential myokines that were up-regulated [84], such as colony-stimulating factor-1 (CSF1), nine that were down-regulated after long-term PA [84], and myostatin [84,85]. Especially, myostatin is interesting in regard to the effects of long-term PA on IR [85]. Although myostatin may not influence insulin-stimulated glucose uptake or phosphorylation of Akt in humans [85], it seems to increase the basal uptake of glucose, glucose oxidation, and lactate production in SkM [85]. Furthermore, this effect is additive to insulin-stimulated glucose uptake in SkM [85].

### 2.2. Adipose Tissue

Most research on human IR now points to the “lipid overflow hypothesis”, which refers to too much (ectopic) lipids being stored in the SkM and liver. This is also highly related to adipose tissue. Adipose tissue expands with chronic overnutrition, but the storage capacity of adipose tissue seems to be limited. Studies on persons with lipodystrophy (lack of adipocytes) have revealed large amounts of fat in the SkM and liver associated with substantial IR [96,97]. In contrast, some studies indicate the existence of “the healthy obese” phenotype, which refers to people that have low IR despite obesity, probably related to the degree of physical activity and/or high storage capacity in white adipose tissue and correspondingly low levels of SkM and liver fat content [9]. Thus, the adipose tissue may be the main orchestrator of lipid availability for SkM and the liver. Studies have revealed that expanding adipose tissue may lead to hypoxia and macrophage and immune cell infiltration that may enhance lipolysis and adipose tissue IR [13,14]. The increased flux of fatty acids and glycerol, and altered release of adipokines and inflammatory mediators from the adipose tissue to the SkM and liver are potent drivers of IR in these tissues (e.g., through *sn*1,2-DAG activation of PKC theta/epsilon), gluconeogenesis, and hyperglycemia [13,14].

Whereas SkM is extensively studied in regard to PA [84], adipose tissue has had substantially less attention concerning this aspect. Adipose tissue inflammation seems to be the main driver of systemic IR via altered adipokine secretion and increased release of free fatty acids and glycerol [9,98]. These factors may promote ectopic lipid deposition and IR in SkM and the liver [9,98]. Some researchers even think that adipose tissue inflammation may be the first step towards developing IR and T2DM in humans [9,98]. Still, few reports are available on the long-term effects of PA on adipose tissue.

#### 2.2.1. Adipose Tissue Inflammation

We found less subcutaneous adipose tissue and, even more pronounced, less visceral adipose tissue mass after long-term PA in the MyoGlu study [99]. Furthermore, long-term PA was also associated with reduced expression of genes coding for adipose tissue macrophage and leucocyte-infiltration, especially in persons at risk of developing T2DM [99]. Reduced expression of genes coding for adipose tissue macrophage-infiltration was strongly correlated to reduced IR assessed by the gold standard euglycemic hyperinsulinemic clamp [99]. Interestingly, reduced adipose tissue inflammation was observed after long-term PA, but not after weight loss from energy restriction [100]. Other studies have also implied that long-term PA may reduce the plasma concentration of several markers of inflammation like C-reactive protein (CRP), IL-6, TNF-α, IL-18, interferon-gamma (INF-γ), and IL-10 in healthy subjects as well as patients with T2DM [94,101,102]. Reduced plasma concentrations of inflammatory mediators after long-term PA may also occur without reduced total fat mass [95]. The observation that long-term PA may reduce immune cell infiltration in adipose tissue is interesting because it may reduce adipocyte IR and thereby reduce lipolysis so that less free fatty acids and glycerol must be taken up into the SkM and liver [71].

#### 2.2.2. Adipokines

The adipose tissue produces and releases a plethora of signaling molecules known as adipokines. Some studies have assessed the effects of PA on adipokines [103,104,105,106]. However, the majority of studies on adipokines and PA are either limited to plasma analyses of one or a few factors (most often leptin and adiponectin) and effects of acute PA [105,106,107,108,109], PA confounded by weight loss [110], or only focus on visceral adipose tissue [103,104,105]. One study analysed gene expression in subcutaneous adipose tissue before and after long-term PA in insulin-sensitive women [109,111]. We published a recent report where we monitored the effects of long-term PA on subcutaneous adipose tissue expression of adipokines [99]. Our study reported that long-term PA affected several potential adipokines, especially for subjects at risk of developing T2DM. After long-term PA, expression of these factors was normalized as compared to healthy control persons [99]. These human results are in line with rodent studies transplanting subcutaneous white adipose tissue from mice after 11 days of cage wheel training into sedentary mice to increase glucose tolerance [112]. The effect was present until day 9 and associated with increased SkM and interscapular brown adipose tissue glucose uptake [112] and alterations in >250 putative adipokines [112]. In humans, such mediating factors may include the soluble leptin receptor [113] and secreted frizzled-related protein 4 (SFRP4) [99].

#### 2.2.3. Adipocyte Mitochondrial Dysfunction

A link between mitochondrial dysfunction and obesity has been proposed [114,115,116,117] based on studies showing that increased visceral adiposity from pre- to postmenopause increases the risk of metabolic diseases [118], and that adipocyte mitochondrial gene expression, including ESR1 (a gene encoding the estrogen receptor α (ERα)), may differ between monozygotic twins discordant for obesity [114]. Deletion of ERα from adipocytes promotes larger adipocytes and the development of obesity in both male and female rodents [119,120]. ERα may regulate mitochondrial function and energy homeostasis in both white and brown adipose tissue via DNA polymerase subunit gamma 1 (Polg1) mitochondrial DNA replication and fission-fusion-mitophagy in mice as well as human [121,122]. ERα is a promising therapeutic target to combat obesity and metabolic dysfunction. Interestingly, one effect of long-term PA is increased ERα action and mitochondrial content in adipose tissue from both mice and human, which is strongly associated with reduced IR [76].

### 2.3. Liver

The liver is inherently difficult to study in humans for obvious reasons; most data are based on imaging studies and reports on indirect measures such as blood markers.

#### 2.3.1. Liver Fat Content

It might seem that long-term PA has a distinct and pronounced effect on reducing liver fat content, as measured by MRS or computer tomography scans [28,123,124]. Several studies have shown a clear trend towards lower ectopic fat content in the liver after long-term PA, especially in subjects with overweight, obesity, and/or non-alcoholic fatty liver disease [125,126]. It may also seem that plasma free fatty acids, triglycerides, total and low-density cholesterol levels, are lower after long-term PA, probably due to reduced liver fat content, in these individuals [125,126]. Furthermore, we have shown that fat content in the liver may be reduced after long-term PA also for normal-weight [26]. The effects of long-term PA on the liver are not only related to weight loss but may exhibit some specificity for PA [28,123,124].

#### 2.3.2. Hepatokines

The liver produces and secretes proteins known as hepatokines that may influence systemic IR. An interesting hepatokine with regards to IR is fetuin-A [127,128,129,130]. Fetuin-A is also linked to diabetes microvascular complications [131] and the development of nonalcoholic fatty liver disease (NAFLD) [132]. Saturated fatty acids may induce IR either by providing substrate for ceramide biosynthesis [133], or by activating toll-like receptor 4 (TLR4) signaling [123]. Saturated fatty acids can bind to fetuin-A, activate TLR4 signaling, and induce adipocyte IR [124]. Moreover, fetuin-A may by itself impair insulin receptor tyrosine kinase activity and insulin action [134,135,136]. Mice lacking fetuin-A are protected from having diet-induced IR [137]. Fetuin-A may be involved in cross-talk between adipose tissue and the liver because fetuin-A can induce the release of adipokines such as IL-6 and TNFα, which in turn can influence hepatocellular signaling pathways [138]. We have shown that long-term PA may decrease fetuin-A levels [130], which correlated to a reduction in liver fat content, adipose tissue inflammation, and IR [127,128,129,130].

### 2.4. Sex Differences

Sex differences in susceptibility to obesity, insulin resistance, and other cardiometabolic traits have been thoroughly described, with pre-menopausal women generally exhibiting beneficial metabolic profiles and less insulin resistance [139]. Men generally have more visceral and hepatic adipose tissue, whereas women have more peripheral and subcutaneous adipose tissue promoting a more insulin-sensitive environment in women than men [140]. Both women and men with T2D have lower peak oxygen consumption (VO_2_max) than non-diabetic controls [141]. Differences in VO_2_max between T2D subjects and controls are larger in women (24%) as compared to men (16%) [141]. We have recently shown large sex-differences in many metabolic traits including mitochondrial functions in adipose tissue [117] in mice. Women rely more on fat oxidation, and less on carbohydrate and protein oxidation during exercise, than men [142]. Moreover, high-intensity interval training (HIIT) increases aerobic capacity in both men and women, although its ability to enhance insulin sensitivity appears to be blunted in women [143].

### 2.5. IR in Pregnancy

IR increases during pregnancy, and if the insulin secretion from the pancreatic β-cells is not able to compensate, hyperglycaemia and eventually gestational diabetes (GDM) may ensue [144]. Lifestyle-based strategies including PA to prevent and treat GDM are cornerstones in management [145]. While exercise have well-documented effects in lowering fasting and postprandial blood glucose concentrations in the mother, the benefits for the infant have yet to be proved [146,147].

### 2.6. Amino Acids and IR

Although most studies have focused on lipids in terms of IR [9], a rising number of studies are now showing potential roles for different amino acids in IR, such as the sulfur amino acids (SAAs) [148] and branched-chain amino acids (BCAA) [149].

#### 2.6.1. SAA

SAAs compromise both semi- and essential amino acids involved in the transsulfuration and glutathione synthesis pathway [139]. Most research so far has focused on SAA and obesity, which is particularly relevant for plasma total cysteine (tCys) (reviewed in [150]). Plasma tCys has been shown to be strongly and positively associated with fat mass and adiposity in several studies [150,151,152]. Following gastric bypass surgery, plasma tCys concentration remains unchanged [153] and predicts weight regain [154]. In addition, in vitro studies suggest that administration of cysteine or the cysteine-cysteine disulfide induces adipose tissue differentiation [152,155]. Other SAAs, such as S-adenosylmethionine and methionine are strongly related to increased fat mass [156] and liver fat [157], respectively.

Concerning IR and T2DM, human studies are limited, perhaps overshadowed by the strong focus on BCAA in T2DM [158]. However, high absolute dietary intake of SAAs and high protein density of particularly cysteine were strongly associated with diabetes-related mortality in the NHANES III cohort [159], and plasma concentrations of total cysteine were positively associated with IR in Hispanic children and adolescents [160]. For the other SAAs, the evidence is less clear, and plasma concentrations of methionine showed no associations with IR in a Japanese cohort [161], whereas a human kinetic study showed decreased hepatic transmethylation reactions and elevated homocysteine in individuals with T2DM, indicating that SAA metabolism is compromised in IR [162]. In animal studies, dietary restriction of SAAs leads to distinct metabolic benefits including reduced IR, decreased adiposity, and decreased oxidative stress [163,164,165,166].

Taken together, several lines of evidence suggest that plasma and dietary SAA, and their metabolism, may be implicated in IR and act as risk factors for T2DM. However, from human data the direction of causality is difficult to establish, especially considering that parts of SAA metabolism are regulated by insulin [162,167]. Moreover, short-term pilot studies on dietary SAA restriction do not seem to affect plasma tCys concentrations or markers of IR to a meaningful degree [168,169,170], although one study reported a decrease in liver fat after 16 weeks of SAA restriction [170].

In contrast, PA has acute [171] and long-term effects on plasma concentrations of SAA [168]. We recently demonstrated that a 12-week combined endurance and strength exercise intervention led to marked changes in plasma SAA profiles of sedentary men [168], with reductions in plasma tCys, which otherwise has proven difficult to attain by dietary intervention in humans. Exploratory analyses showed that a more marked tCys reduction was correlated with reduced IR. Moreover, reductions in plasma tCys levels are associated with the expression of genes involved in mitochondrial metabolism in both SkM and adipose tissue. We also found signs of lower glutathione synthesis [168], corresponding to lower “low-grade” inflammation after long-term PA [99]. This is suggestive of a joint effect of long-term PA on SAAs, and particularly tCys, paralleling reduced IR that so far has not been shown in dietary studies.

#### 2.6.2. BCAA

Accumulation of leucine, isoleucine and valine in blood had already been linked to obesity by the 1960s [172], but was later largely ignored. It was not until the technological progress of large-scale metabolomic methods in the early 2000s that large prospective studies identified that initial blood BCAAs levels could predict future T2DM and IR [172,173,174,175,176,177,178,179,180]. These observations sparked much research on BCAAs and IR [140]. Many reports have now shown elevated blood levels of BCAAs in IR of normal weight and obese men and women [140,172,176,181,182,183,184,185].

Several processes may lead to elevated blood BCAAs [186], such as the diet, repressed BCAAs catabolism in adipose tissue, liver, and SkM [186,187,188,189,190,191,192], an altered balance between protein synthesis and proteolysis in SkM and splanchnic tissue (gastrointestinal tract, liver and other visceral organs) [193], altered gut microbiota [194], certain genetic single nucleotide polymorphisms [191], and the amount of urinary excretion of BCAAs [195].

In adipose tissue, liver, and SkM, blood BCAAs are taken up by the large amino acid transporter (LAT1) [196,197]. BCAAs are then catabolized by a series of 43 enzymes mostly located inside mitochondria [187]. BCAA catabolism is essential in BCAA homeostasis in humans [190]. The rate-limiting step in BCAAs catabolism is the branched-chain alpha-keto acid dehydrogenase (BCKDH) [187]. The highest BCKDH activity, and thus BCAAs’ catabolism, occurs in adipose tissue, followed by liver and SkM [187,188,189]. After oxidation by BCKDH, BCAAs are trapped within mitochondria, except the intermediate metabolite 3-hydroxy-isobutyrate (3-HIB) [198]. 3-HIB may thus serve as a blood marker of valine degradation [187] and is highly related to IR [198]. The remaining carbons from BCAAs’ catabolism are then oxidized by the citric acid cycle [187].

However, although it might be tempting to speculate about a causal role for blood BCAAs on IR, we [185] and others [195] have shown that long-term PA does not alter blood BCAA levels, despite substantially reduced IR. Instead, we found that the expression of genes involved in both adipose tissue and SkM BCAAs catabolism increased substantially after long-term PA, and BCAA catabolism may mediate some of the effects of PA on reduced peripheral IR [185]. This is in line with rodent reports showing that repressed BCAAs catabolism can induce IR and that restored flux through the BCAAs’ catabolism pathway reduces IR [199].

## 3. Summary

Long-term PA may counteract IR via a complex interplay between different organs, such as the liver, SkM, and adipose tissue. Indirect mechanisms, such as increased energy consumption and improved body composition, reduced adipose tissue inflammation and ectopic lipid deposition in the liver, and enhanced metabolic capacity of the SkM, may subsequently lead to direct effects on insulin signaling in target tissues.

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
