# Peer review of "Potential Mechanisms for How Long-Term Physical Activity May Reduce Insulin Resistance"

_metabolites, 2022, doi:10.3390/metabo12030208_

Round 1
Reviewer 1 Report
This MS is a very well written piece of work, making an update of a very relevant issue, I dare say, an every day one, with an interest to the big medical/scientific community, not just the specialized one. We know of the relevance of physical activity, but a careful inspection of how it works makes the effort of exercising worthy.
Major comments:
What is to improve a resistance? To make it more resistant. When the desired effect is to diminish insulin resistance or, the other way round, to improve insulin sensitivity. In this regard, this expression is vastly used throughout the MS (either improve, enhance, or equivalent wording). The MS should be extensively revised from the very title.
Moreover, I regret there is no consideration of endocrinology –the cross-tissue interaction described with whole sections addressing myokines and adipokines- whereas there is a pretentious inclusion of the signaling pathways involved (in expressions such as signaling molecules or molecular mechanisms), since the description of them is very scarce. This comment does not point to the contents but the allusion of them. A little touch to refocus the presentation should be more than enough to attend this issue. This includes title and abstract.
Minor comments:
Going into minor stuff, there are many typing errors that need to be changed –a spelling corrector would help- plus some grammatical errors (no correlation between subject and verb in number in sentences or discordance between subject and verb in a sentence) and some word repetitions in the same sentence or very close nearby.
Another is the use of e.g. This abbreviated form is repeatedly used, sometimes does not seem to fit. For example, at the beginning of a sentence.
IR and PA should also be defined in the text, upon first use (not only in the abstract). The same holds for SkM. In this sense, perhaps adding a brief definition (half a line) of some specific words or initials (e.g. Polg1 or SFRP4), would help an ampler public.
L40: if you are not to address the subclassification of T2DM (this is ok), AND may not be the proper connector. Rather use sth like “for which”.
L55: “blunted, but increased (insulin secretory) response” This sounds contradictory.
L59. “in [11],” better if introduced or described in some way.
L 68: “Both increased physical activity (PA) and dietary energy restriction may prevent T2DM, reduce morbidity and mortality, and even reverse T2DM [15-18].” Guess items should be ordered in degree of importance. Sth like “Both increased physical activity (PA) and dietary energy restriction may prevent or even reverse T2DM, reduce morbidity and mortality [15-18].” seems more appropriate. Whereas PA should be defined earlier (L67).
L77. “SkM contraction may increase glucose uptake 50-fold and is preserved in IR SkM through distal insulin signaling effectors different from those activated by insulin [22,23).” Unsuitable phrasing.
L80. Basal IR needs to be explained, as some readers may not be familiarized with the concept.
L97. IR cannot be measured, but determined, estimated, or sth alike. You measure blood glucose levels or insulin ones.
L121. “For endurance training, in week 1-3 cycle intervals …” 1-3 is not clear, and not consistent with the lines before. You should use comma to separate week 1 from the 3 cycles.
L133. “which still persist ..:” it is not clear what persists.
L162-165. Whole sentence should be reformulated (rewritten).
L212. mitochondrial volume density deserves to be explained, as it a rather non-intuitive concept. L213. “both increased mitochondrial length and cross sectional area”. These two parameters address mitochondrial size but not its number (also included in volume density). Perhaps this can be better discussed.
L328. “In regards to subcutaneous adipose tissue, some researchers have studied insulin sensitive women”. This sounds awkward, better if rephrased.
L180 & 183. Food should be in singular.
Please find attached your MS, where I have highlighted serveral minor errors (typing or verb concordance).

Reviewer 2 Report
Lee-Ødegård et al., present a very nice review focused on potential mechanisms for insulin resistance. The authors reviewed the literature to demonstrate that insulin resistance can be improved by physical activity. Moreover, the authors nicely covered 3 main metabolic organs of the body (skeletal muscle, adipose tissue and liver) to have an integrated view of insulin resistance.
This review is very well written (congratulations to the authors). Two aspects were not covered by the authors and that reduce the impact of this review article:
1) potential differences between male and women with IR (eg. Estrogen or Testosterone?)
2) late pregnancy is a state of insulin resistance. It would be interesting if the authors can write a section about that and how that impact on fetal outcomes. Some literature is available comparing sedentary women versus active women.
Aside from that, I think the review is good enough to be published.
Minor comments:
Line 258: “acute exericse [83]” (typo)
Line 266: “exercise” (typo)
Line 365: “haveshown” (typo)
Reviewer 3 Report
Interesting and well-written article, which needs few more additions to make it more comprehensive and also more updated.
Many references included in this article are more than 20 years old. By contrast, only 4 studies published in 2021 are listed, and none from 2022. This has to be significantly improved - please see my comments below.
- Section 1.1 IR. Please briefly mention the role of skeletal muscle in insulin resistance. Recent publications:
Diabetes Metab J. 2022 Jan;46(1):15-37. doi: 10.4093/dmj.2021.0280.
Int J Mol Sci. 2021 Aug 28;22(17):9327. doi: 10.3390/ijms22179327.
2. Section 1.2. PA vs. diet. Please briefly mention the role of low carbohydrate diets on controlling the overall cardiometabolic risk, which is even more important in the adolescents.
Recent publications:
Nutrients. 2021 Sep 28;13(10):3422. doi: 10.3390/nu13103422.
Exp Ther Med. 2021 Jan;21(1):90. doi: 10.3892/etm.2020.9522.
3. Section 2.1.2. Adipokines. Please briefly mention the role of resistin associated to physical activity. Very recent publication:
Int J Sport Nutr Exerc Metab. 2021 Dec 1:1-9. doi: 10.1123/ijsnem.2021-0148. Online ahead of print.
This has a particular importance since resistin is well known to be a key adipokine for insulin resistance, diabetes and cardiovascular diseases: Curr Pharm Des. 2014;20(31):4961-9. doi: 10.2174/1381612819666131206103102.
4. Section 2.3.2. Hepatokines. Please briefly mention the role of Fetuin-A on cardiometabolic diseases.
Recent publications:
Cardiovasc Diabetol. 2022 Jan 8;21(1):6. doi: 10.1186/s12933-021-01439-8.
Int J Mol Sci. 2021 Jun 21;22(12):6627. doi: 10.3390/ijms22126627.
Round 2
Reviewer 3 Report
no further comments